# Three-Dimensional Polymer Variable Optical Attenuator Based on Vertical Multimode Interference with Graphene Heater

**DOI:** 10.3390/mi13122116

**Published:** 2022-11-30

**Authors:** Xinru Xu, Yuexin Yin, Mengke Yao, Xiaojie Yin, Feifei Gao, Yuanda Wu, Changming Chen, Fei Wang, Daming Zhang

**Affiliations:** 1State Key Laboratory of Integrated Optoelectronics, College of Electronic Science and Engineering, Jilin University, Changchun 130012, China; 2State Key Laboratory of Integrated Optoelectronics, Institute of Semiconductors, Chinese Academy of Sciences, Beijing 100083, China; 3Shijia Photons Technology, Hebi 458030, China; 4College of Materials Science and Opto-Electronic Technology, University of Chinese Academy of Sciences, Beijing 100049, China

**Keywords:** optical devices, multimode interference, variable optical attenuator, 3D integration

## Abstract

Low-power-consumption optical devices are crucial for large-scale photonic integrated circuits (PICs). In this paper, a three-dimensional (3D) polymer variable optical attenuator (VOA) is proposed. For monolithic integration of silica and polymer-based planar lightwave circuits (PLCs), the vertical VOA is inserted between silica-based waveguides. Optical and thermal analyses are performed through the beam propagation method (BPM) and finite-element method (FEM), respectively. A compact size of 3092 μm × 4 μm × 7 μm is achieved with a vertical multimode interference (MMI) structure. The proposed VOA shows an insertion loss (IL) of 0.58 dB and an extinction ratio (ER) of 21.18 dB. Replacing the graphene heater with an aluminum (Al) electrode, the power consumption is decreased from 29.90 mW to 21.25 mW. The rise and fall time are improved to 353.85 μs and 192.87 μs, respectively. The compact and high-performance VOA shows great potential for a variety of applications, including optical communications, integrated optics, and optical interconnections.

## 1. Introduction

To meet ever-increasing demands for high-speed and large-capacity communication, wavelength division multiplexing (WDM) has been introduced into metro and access networks. Array waveguide grating (AWG) is a suitable structure for a large port count on-chip WDM systems [1,2]. The channel count of AWG has reached 512 [3]. Wavelength manipulation devices are always combined with optical power attenuators (VOA), making it possible to selectively block or equalize one or more of the transmitted channels dynamically. For an on-chip WDM system, the monolithic integration of AWG and a variable optical attenuator (VOA) array is an effective method to preserve the power uniformities across all signal channels, thereby deteriorating the optical signal-to-noise ratio (OSNR) and bit error rate (BER) [4,5,6,7]. Low propagation loss and high coupling efficiency contribute to silica-based planar lightwave circuits (PLCs) being commercially available. However, the low thermal optical coefficient (TOC) of silica leads to large power consumption for VOAs and optical switches [8,9]. Compared with silica, polymer-based PLCs show low power consumption due to a higher-order TOC. Moreover, low-cost and simple fabrication make polymer-based PLC an ideal platform for monolithic hybrid integration. In this paper, we propose a polymer three-dimensional (3D) VOA based on vertical multimode interference (MMI) at the O-band. The 3D hybrid integration VOA is inserted into silica-based PLC through a compatible fabrication process wherein the polymer material is filled into a silica groove. Because of the high TOC of polymers and the large thermal conductivity of silica, the VOA has both a low power consumption and a fast response speed [10,11,12]. To cooperate with the temperature-sensitive AWG, both the structure and electrode of VOA are analyzed well and optimized for a compact structure and low power consumption. While the temperature change is 7 K, the proposed VOA shows an insertion loss (IL) of 21.18 dB at 1310 nm with a length of 3092 μm. A power consumption of 29.9 mW and response times of 677.70 μs and 244.73 μs are achieved while the aluminum (Al) electron is applied on poly-methymethacrylat (PMMA) upper cladding.

Due to its high electrical conductivity and optical transparency, graphene, as a new type of two-dimensional material, has been widely researched and applied in various applications, including absorbing modulators [13,14,15], low-power-consumption optical switches [16,17,18,19,20], photodetectors [21,22,23], and optical sensors [24,25,26,27]. For graphene-based thermos-optic (TO) switches, graphene microheaters could be attached to the top of the devices with negligible absorption for the light-polarized perpendicular [28]. With graphene microheaters introduced, the power consumption of the VOA proposed in this paper is decreased to 21.25 mW. The rise and fall time are improved to 353.85 μs and 192.87 μs. 

As a proof-of-concept, the proposed structure offers an opportunity for silica and polymer monolithic hybrid integration. The polymer PLC provides several functions in which silica is not used, including electro-optic modulation [29], an optical amplifier [30], and a biological sensor [31]. The concept provides a low-power-consumption hybrid and multifunctional and large-scale integration platform.

## 2. Device Design and Simulation

The schematic of the proposed MMI-based VOA is shown in Figure 1. A polymer core is inserted between silica waveguides. After silica-based PLC device fabrication, we etch the waveguide to form grooves for polymer material. The metal electrode will be formed above the polymer cladding. The light from silica waveguides will be coupled into silica output waveguide through polymer MMI. With electric power applied to the electrode, a phase shift will be introduced by the temperature changing of polymer MMI. The position of self-imaging is changed. Therefore, the light from the silica output waveguide is attenuated.

The cross sections of the silica waveguide and polymer MMI are shown in Figure 2a,b, respectively. Silica-based devices are fabricated on 2%-Δ platform. Both of width (W_silica_) and thickness (H_silica_) of silica waveguides are 4 μm to realize single-mode propagation [32,33]. For a compact size, the gap between polymer cladding and silica core (H_gap_) is 4 μm. A 4 μm thick polymer layer (H_polymer_core_) is left above the silica cladding. The width W_MMI_ of the polymer MMI is 12 μm. There is a 7 μm thick polymer cladding (H_polymer_clad_) above the silica cladding to protect the polymer MMI. The refractive indices at 1310 nm of the polymer core and cladding are 1.5812 and 1.4769, respectively. 

According to the principle of self-imaging in an MMI structure, the beat length of an MMI can be written as [34]
(1)$$L_{\pi }=\frac{\pi }{\beta_{0}-\beta_{1}}$$
where *β*_0_ and *β*_1_ are the propagation constants of the fundamental and first-order modes in the MMI section, respectively. For the general interference type, a self-image is reproduced at the position of $L_{MMI}$ [34]
(2)$$L_{MMI}=p(3L_{\pi })$$
where p is an integer (p=2,4,6…).

In this paper, a free space wavelength of 1310 nm is considered. Due to the low width of the MMI, the calculated self-imaging lengths for transverse-electric (TE) and transverse-magnetic (TM) polarizations are similar. To cooperate with the design of silica-based PLC devices, the VOA is designed for TE polarizations. A 3D semivectorial beam propagation method (BPM) is used to simulate light propagation. As shown in Figure 3, the calculated self-imaging lengths are 1547.5 μm, 3092 μm, 4642.5 μm, and 6191.5 μm, respectively. The IL increases while the length of MMI increases. On the other hand, the phase shift caused by the thermo-optical effect will increase and lower the power applied on the electrode when the length of MMI is increasing.

Figure 4 shows the relationship between IL and temperature with different MMI lengths. With the length of MMI increasing, the off-temperature will be lower. The maximum attenuator happens when the temperature changes are 11 K, 7 K, 5.4 K, and 4.6 K, respectively. To keep the trade-off between size and power consumption, we chose two and three for the period lengths of the MMI.

The normalized transmissions under on- and off-temperature changes with the length of 3092 μm and 4642.5 μm are shown in Figure 5. While the length is 3092 μm, the maximum attenuation of 41.89 dB happens at 1316 nm. While the temperature changes are 1.3 K and 7 K, the ILs are 0.58 dB and 21.18 dB at 1310 nm, respectively. While the length is increased to 4642.5 μm, the phase shift introduced is larger with the same temperature change. Therefore, the off-temperature change is changed to 5.4 K. The maximum attenuation of 38.05 dB is obtained at 1315 nm. While the temperature changes are 1.3 K and 5.4 K, the ILs are 0.60 dB and 21.20 dB at 1310 nm, respectively. The 1 dB bandwidths of 3092 μm and 4642.5 μm are 27 nm and 19 nm, respectively. Though increasing the length of MMI will lower the off-temperature, the crosstalk and loss will be worse. According to the results of the simulation, the 3092 μm long VOA is more practical for applications. To cooperate with the CWDM system in the 100 nm wavelength range, lengths of VOAs for central wavelengths, 1270 nm, 1290 nm, 1310 nm, and 1330 nm, are optimized to 3184 μm, 3146 μm, 3092 μm, and 3062 μm, respectively.

## 3. Thermal Field Analysis and Optimization

To determine the modulation efficiency of VOA, thermal field distribution is calculated through the general heat-diffusion equation
(3)$$\rho \left(T\right)C_{p}\left(T\right)\frac{\partial T}{\partial t}=\nabla [k(T)\nabla T]+H(x,y,z,t)$$
where *ρ* is the material density, *C_p_* is the specific heat capacity, *T* is the temperature, *k* is the thermal conductivity, and *H* is the heat source distribution, which corresponds to the heat source. The variables *x*, *y,* and *z* represent the axes of the coordinates system, and *t* is time.

The heat-diffusion equation was solved numerically using COMSOL Multiphysics. The smallest size of mesh cells is 0.35 nm. Figure 6 shows the thermal field distribution of the VOA cross-section when the aluminum (Al) electrode heats the VOA. As shown in Figure 6a, the temperature change weakens as the depth of VOA increases. To investigate further, the temperature distribution along Y planes is extracted and plotted in Figure 4b. We analyzed the relationship between temperature and Y planes according to linear fitting. When the height position is above zero, the slope is 1.77. While the height position is less than 0, the slope is 0.44. Therefore, the depth of 4 μm is chosen to determine the thermal variation of the VOA.

Since most optical components such as MMIs, directional couplers, and Y-branches are realized using horizontally placed planar lightwave circuits, the side electrode proves most suitable when an index gradient in the horizontal direction is needed to drive the devices [35,36]. The slight index change in the Y direction does not influence the mode profile significantly for either the TE or TM mode. In this paper, we design a polymer VOA based on vertical MMI, so we need a vertical temperature gradient to drive the device. Considering fabrication requirements and power consumption, we use a 6 μm wide electrode to simulate the thermal field. At the same heating temperature, we calculate the temperature gradients in different axial directions of the VOA. Figure 7a shows the temperature change in the X direction of the top electrode and the side electrode, respectively, under the same heating conditions. We can find that when the top electrode heats the VOA, the temperature gradient of the waveguide in the X direction changes less than 0.5 K and is symmetric. When the side electrode heats the VOA, the temperature gradient of the waveguide in the X direction changes by 2 K. The small temperature difference in the X direction has little effect on the change of the waveguide mode. To investigate further, the temperature distribution along the Y planes is extracted and plotted in Figure 7b. We found that the VOA has a larger temperature gradient in the Y direction when the top electrode is heated. Therefore, under the same conditions, the VOA under the heating of the top electrode is more likely to have a refractive index difference in the vertical direction. Figure 7c is the heating process under the directional pulse voltage. Therefore, using the top heater is a more efficient method. In summary, the top heater can generate the temperature gradient effectively in the vertical direction and has a high heating rate while maintaining a uniform temperature profile in the horizontal direction.

To further reduce power consumption, we introduce graphene into the electrode structure. In the traditional structure, we need an upper cladding layer to avoid metal absorption. The upper cladding is the main reason for the low heating efficiency and slow heating speed. We remove the upper cladding and use the high thermal conductivity of graphene (thermal conductivity of graphene layer kg = 5300 W/mK; thermal conductivity of polymer kp = 0.19 W/mK) to assist the conduction of heat produced by the electrode [16,19]. According to the conclusions of the previous section, the heating efficiency is higher when the electrode is directly above the VOA. In consideration of process tolerances and the absorption of light by metals, we placed the Al electrodes 2 μm away from the VOA. Figure 8a shows the cross-section of VOA with graphene electrodes. Figure 8b,c show the light field distribution and thermal field distribution of the VOA, respectively. Graphene can induce large absorption loss to the tangential electric field while causing no loss to the normal electric field. The polymer waveguide we designed has a large cross-sectional square core and strong asymmetry in the vertical direction here, achieving negligible light absorption for both TE and TM polarizations. We find that graphene can effectively assist the conduction of heat produced by the electrode and does not affect the optical field distribution of VOA, as shown in Figure 8b.

To calculate the power consumption and response time, we use the multiphysics field combined with the electric field and the thermal field by COMSOL. First, we calculate the model with an Al electrode above the VOA directly with PMMA upper cladding. The width and length of the Al electrode are 3092 μm and 6 μm, respectively. We apply a pulsed current to the electrodes and use the Joule heating effect to heat the VOA. By detecting the voltage across the electrodes, the final power applied to the electrodes is determined. Electric heating will represent the actual situation better than the heat source power used in the traditional simulation. Figure 9 shows the response time of the VOA with the Al electrode heated. The frequency of the signal is 500 Hz. When the input power is 29.8 mW, the temperature of the modulation area rises by 7 K, and the VOA achieves the attenuation function. The rise time is 677.70 μs, and the fall time is 244.73 μs. The reason for the slow rise time is that the temperature change of the electrode is also not instantaneous when the electrode’s heating is taken into account. The inset shows the electrode temperature rise. Then, we calculated the graphene electrodes mentioned in the previous section. As shown in Figure 9b, when the input power is 21.25 mW, the temperature of the modulation area rises by 7 K, and the VOA achieves the attenuation function. The rise time is 353.85 μs, and the fall time is 192.87 μs.

## 4. Discussion

The performance comparison between VOAs on different platforms is listed in Table 1. For low power consumption, several investigations have been developed on silica-based PLC. In [37], an air trench is optimized well and introduced to a four-channel silica VOA array. According to the experimental result, the power consumption was successfully reduced to as low as 155 mW at an attenuation of 30 dB. With a suspended narrow ridge structure introduced, an ultralow power consumption of 20 mW is obtained based on a silica-based PLC platform [7]. However, this structure requires a somewhat-complicated fabrication process, including twice-dry etching and once-wet etching. The silicon-on-insulator (SOI) platform is attractive for large-scale integration PICs thanks to its compact size, large TOC (∼1.8 × 10^−4^/K), and compatible fabrication with a complementary metal-oxide-semiconductor (CMOS) fabrication technique. In [6], an SOI-based VOA with 900 μm long Ta heaters above the 1200 μm long MZI arms is demonstrated. Low power efficiency contributes to a 15 dB attenuation with a power consumption lower than 35 mW. Another heating method for silicon waveguides is a doped waveguide as a micro-heater. This kind of microheater is close to the waveguide. Therefore, an attenuation of 30 dB at 50 mW power consumption is achieved in [38]. The most complicated problems for SOI VOA are low coupling efficiency and large polarization-dependent loss (PDL), which limits its usage in practical applications. The polymer-based VOA show low power consumption due to a TOC of −1.86 × 10^−4^/K. In [39], a fluorinated polymer VOA is demonstrated, showing a low operating power of less than 30 mW. The multimode waveguide VOA shows an attenuation of >30 dB. The polymer-based VOA exhibits long-term reliability through the test of Telcordia GR1221. A 1 × 2 polymer-based digital optical switch with radiation-type attenuators [40] shows a large extinction ratio (ER) of 45 dB due to a quartz substrate with grid patterns, which is also meaningful for our device. Our previous work in [11], a VOA with an ER of 18.64 dB with a power consumption of 8.72 mW, was achieved on a polymer/silica hybrid waveguide platform. In this paper, we propose a method for achieving monolithic integration of a silica-based and polymer-based PLC device. The vertical VOA exhibits an attenuation of 21.18 dB with 21.25 mW power consumption. This structure provides a large-scale integration of PLC devices.

## 5. Conclusions

In conclusion, we have proposed a polymer VOA based on vertical MMI. The simulation result has shown that the VOA has good performances, such as a low IL, a high extinction ratio, and low power consumption. We not only optimized the traditional electrode-heating position but also proposed a graphene-electrode-heating method. Compared with the traditional top electrode, our proposed method reduces power by 8.55 mW and improves the response time to 353.85 μs. Low-power VOA is more beneficial to integrate with AWG. The footprint of 3092 μm × 4 μm is much smaller due to the vertical MMI structure. The VOA shows great potential in the optical communication field.

The structure of VOA mentioned in this paper is a conceptual structure that can realize low-cost and simple-process 3D photonic integration. The concept not only provides a solution to improve the integration degree but also provides the hybrid integration of various materials, realizing the expansion of on-chip functions.

## Figures and Tables

**Figure 1 micromachines-13-02116-f001:**
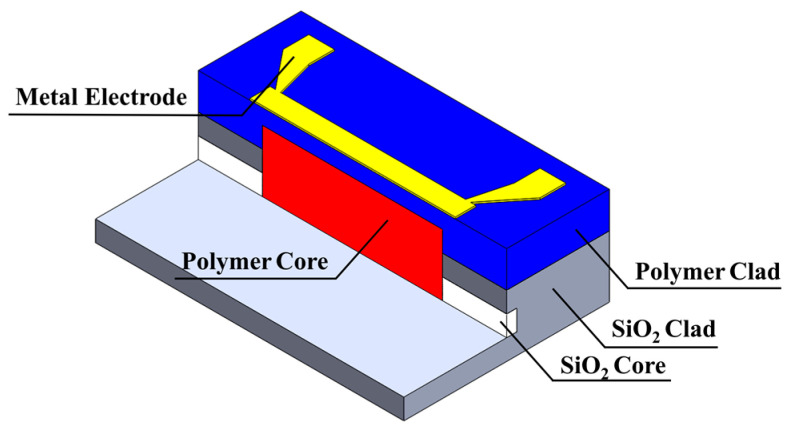
Schematic of the proposed VOA.

**Figure 2 micromachines-13-02116-f002:**
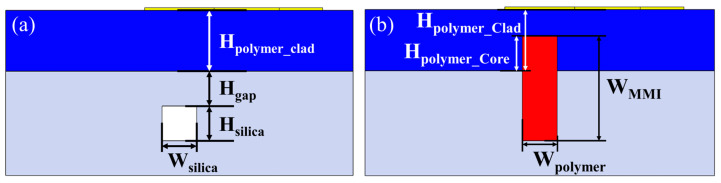
Cross-section of (**a**) silica waveguide and (**b**) polymer MMI.

**Figure 3 micromachines-13-02116-f003:**
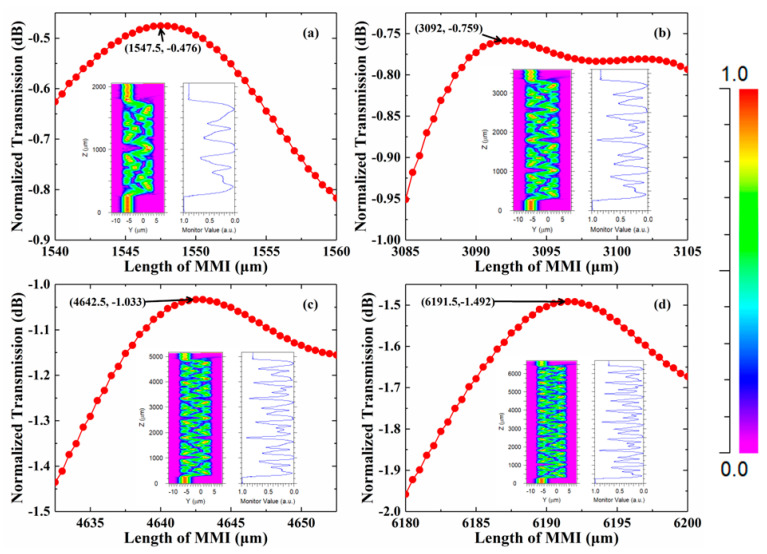
Optimized lengths of MMI that are (**a**) 1-period, (**b**) 2-period, (**c**) 3-period, and (**d**) 4-period. The inserts are light propagation in the MMI.

**Figure 4 micromachines-13-02116-f004:**
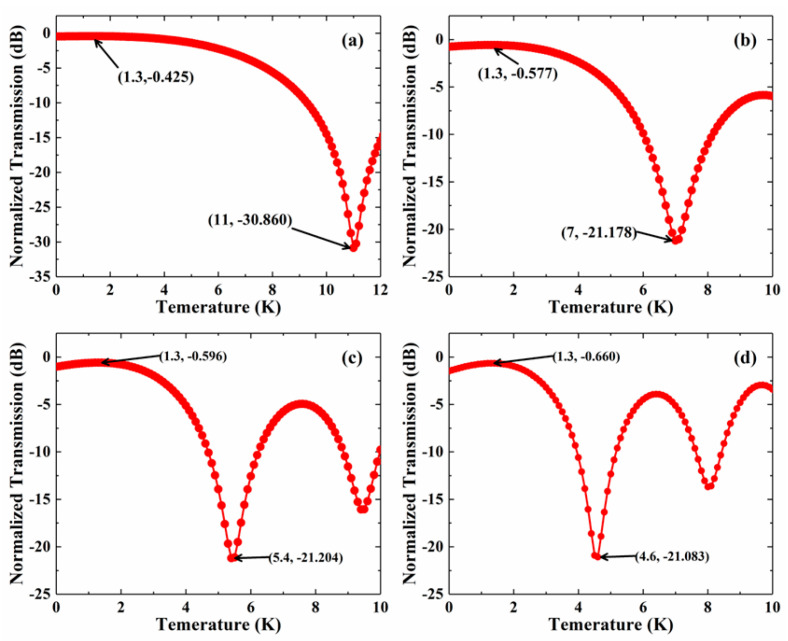
The relationship between IL and temperature change over (**a**) 1-period, (**b**) 2-period, (**c**) 3--period, and (**d**) 4-period-long intervals.

**Figure 5 micromachines-13-02116-f005:**
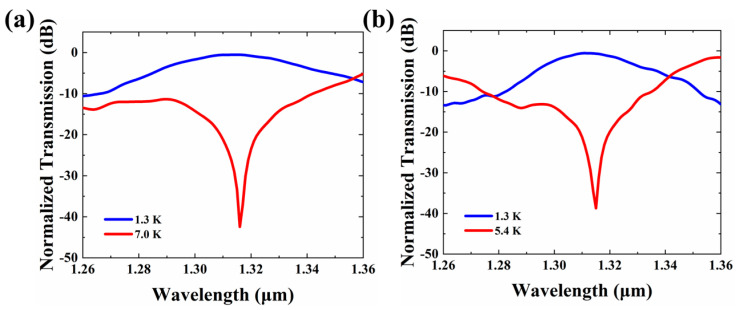
Normalized transmission under on-off voltage temperature with a length of (**a**) 3092 μm and (**b**) 4642.5 μm.

**Figure 6 micromachines-13-02116-f006:**
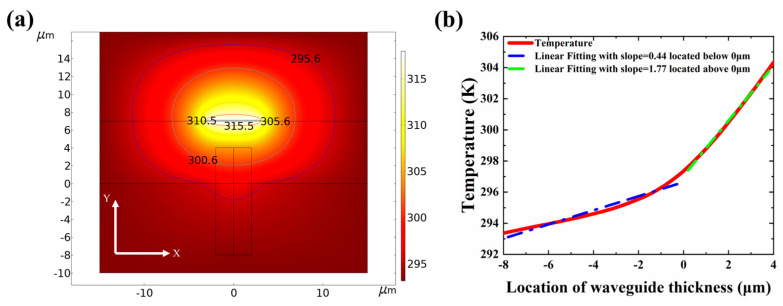
The simulation of the temperature distribution of VOA. (**a**) The thermal field distribution of the VOA. (**b**) The specific value of the change of the temperature in the Y direction with the position distribution.

**Figure 7 micromachines-13-02116-f007:**
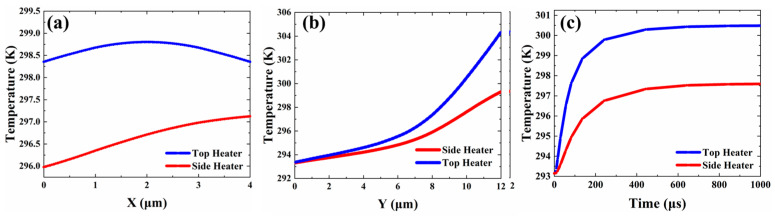
The temperature of the VOA when using different heaters. (**a**) The temperature change in the X direction. (**b**) The temperature change in the Y direction. (**c**) The change of waveguide’s average temperature under pulsed voltage.

**Figure 8 micromachines-13-02116-f008:**
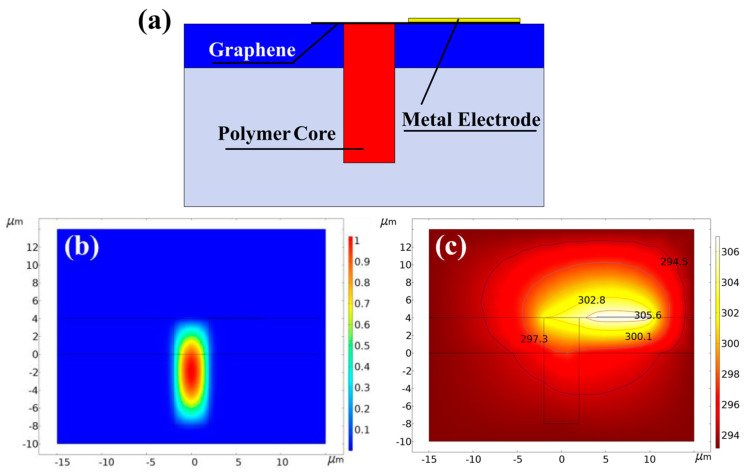
(**a**) The cross-section of VOA with graphene electrodes. (**b**) Optical field distribution in the heating region. (**c**) Thermal field distribution in the heating region.

**Figure 9 micromachines-13-02116-f009:**
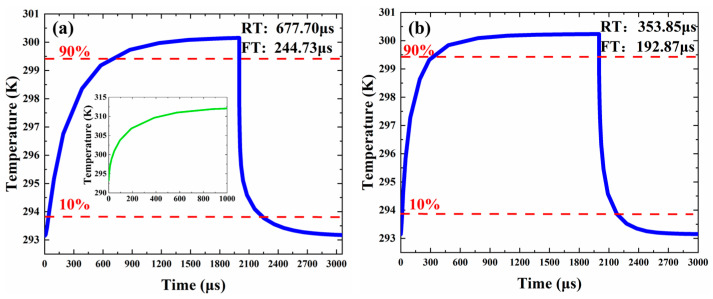
(**a**) The response time of the VOA with the top electrode heated. Inset: The temperature rise of the electrode over time when the input power is 29.9 mW. (**b**) The response time of the VOA with the graphene electrode heated.

**Table 1 micromachines-13-02116-t001:** Comparison of the VOA on different platforms.

Reference	Materials Platform	Attenuation (dB)	PC (mW)
[6]	SOI	15	35
[38]	SOI	30	50
[37]	Silica-on-silicon	30	155
[7]	Silica-on-silicon	40	20
[39]	Polymer	30	30
[40]	Polymer	45	50
[11]	Polymer	18.64	8.72
This work	Polymer	21.18	21.25

## Data Availability

Not applicable.

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
