# Peer review of "Three-Dimensional Polymer Variable Optical Attenuator Based on Vertical Multimode Interference with Graphene Heater"

_micromachines, 2022, doi:10.3390/mi13122116_

Round 1

Reviewer 1 Report

Comments:

In this manuscript, a compact and low power consumption VOA with vertical MMI structure is proposed and well analyzed. The structure is novel. The simulation is abundant and convincing. The current manuscript is well described, scientifically sound and technically interesting. The hybrid integration of silica and polymer is not only promising for optical communication but also biosensor. I recommend it for publication in Micromachines after addressing the following comments.

1.       VOA is a key element in WDM system. Therefore, bandwidth of the VOA is crucial. However, in this article the bandwidth of the VOA is 27 nm only. The method to maintain a large wavelength range should be discussed.

2.       Although the reported work is interesting it lacks some specifics on what type of fabrication process errors it can tolerate. In fact, in such a system it would be of great interest an analysis of the effect of misalignment or fabrication defect of the single element to the whole system. This is probably the main block for developing a technology with 3D integration.

3.       The introduction can be improved. The articles related to the some applications of graphene  materials should be added such as Sensors 2022, 22, 6483; ACS Sustain. Chem. Eng. 2015, 3, 1677–1685; RSC Adv. 2022, 12, 7821–7829; Talanta 2015, 134, 435–442.

4.       The text information of some figures is not clear, and the sum of feet needs to be adjusted, as shown in Figure 2,3,6,9

Reviewer 2 Report

The paper provides theoretical development of three-dimensional polymer VOA based on vertical multimode interference with graphene heater. Compared with aluminum electrode, the power consumption is decreased from 29.90 mW to 21.25 mW by placing the graphene heater in direct contact with the waveguide core. I think there are several points, discussed below, that require further clarity before the paper would be ready for publication.

1. How much loss will be increased when graphene heater directly contacts the waveguide core? The author should provide more references and data support.

2. What was the mesh cells size taken?

3. In my opinion, the authors should classify their results in the discussion even more in comparison to other configurations and/or materials (such as VOA integrated with silicon, III-V semiconductor, and lithium niobate).

Round 2

Reviewer 1 Report

Accept in present form.

Reviewer 2 Report

The authors have addressed the reviewer's comments. So, this paper should be published in micromachines .